# Nematicidal Potential of Thymol against *Meloidogyne javanica* (Treub) Chitwood

**DOI:** 10.3390/plants12091851

**Published:** 2023-04-30

**Authors:** Eleni Nasiou, Ioannis O. Giannakou

**Affiliations:** Laboratory of Agricultural Zoology and Entomology, Department of Science of Crop Production, Agricultural University of Athens, Iera Odos 75, 11855 Athens, Greece

**Keywords:** terpenes, root-knot nematodes, egg mass, sublethal concentration activity, egg differentiation

## Abstract

Root-knot nematodes (RKN; *Meloidogyne* spp.) are obligatory endoparasites with worldwide distribution which cause severe damage to agricultural crops. The present study aimed to define the nematicidal activity of thymol on different life stages of the root-knot nematode *Meloidogyne javanica* (Treub) Chitwood, at concentrations of 37.5–1000 μL/L. This study is the first to report the effect of thymol on egg differentiation and also its vapor and sublethal concentration activities. A mortality of greater than 90% of *M. javanica* second-stage juveniles (J2s) occurred after 96 h of exposure at a concentration of 500 μL/L. At this concentration, thymol inhibited 59.7% of nematode hatching. In addition, the use of thymol at sublethal concentrations reduced the number of females per gram in tomato roots in a pot test, as well as inhibiting egg differentiation. On the contrary, no nematostatic effects were observed in paralysis bioassays. The results presented here indicate that the use of thymol may show its potential as a source of a new sustainable nematicidal product.

## 1. Introduction

Root-knot nematodes (RKN; *Meloidogyne* spp.) are one of the most damaging pests that affect the quantity and quality of many annual and perennial crops, causing major yield losses equivalent to 1 billion euros annually [1,2]. *Meloidogyne* species have an extremely broad host range of over 2000 cultivated plant species, which can cause severe damage to plants via root gall formation and root dysfunction [2,3].

The number of active substances allowed for plant protection decreased under Directive 91/414/EEC (followed by Regulation 1107/2009/EC), which resulted in the ban and withdrawal of a large number of pesticides from the EU market [4]. Moreover, concerns about environmental and health issues have been leading to the removal of most of the synthetic nematicides on which the control of nematodes has relied throughout the past century [5,6], therefore increasing the interest in essential oils (EO) and terpenes as sources of environmentally sustainable biopesticides [7]. An example is methyl bromide, a common fumigant which has a broad spectrum of target pests and has been banned for its harmful effects since 2005 [8]. Controlling nematodes is a challenging task due to several factors including the lack of inherent resistant mechanisms in numerous vegetable crops, the ineffectiveness of preventative strategies against nematode invasion and the limited availability of biological control agents (BCAs) in the market. The withdrawal of key nematicides has increased the need for alternative and safer management methods that are more environmentally friendly and protective of farmers and consumers [9,10,11,12].

Plant-derived extracts, known as botanical pesticides, are considered potential alternatives to synthetic nematicides since they have fewer side effects on non-target organisms and the environment [13,14,15]. Plants synthesize a diversity of secondary metabolites which prominently play a significant role in plant defense against pathogens, herbivores and other environmental stresses [16,17]. Plant secondary metabolites are organic compounds or phytochemicals which consist of three major groups: terpenoids, alkaloids and phenolics [18].

In particular, terpenes, which can be the main volatile constituents of EO, have a wide range of biological activities and play a significant role by providing plants with natural protection against various pathogens and pests, including nematodes [19,20,21,22,23]. One of these terpenes is thymol (C_10_H_14_O; 2-isopropyl-5-methylphenol, IPMP) (Figure 1), a phenolic monoterpene derivative of cymene, which is isomeric with carvacrol, found as the main compound in *Thymus vulgaris* and *Thymus zygis* essential oils [24,25].

Thymol has shown an interesting number of bioactive properties, such as antifungal [26,27], antibacterial [28], acaricidal [29], anthelmintic [30] and nematicidal [22,31,32,33] properties. Oka [34] reported that thymol inhibited the hatching of *Meloidogyne javanica* at concentrations higher than 125 μL/L in in vitro conditions. Moreover, Eden^®^ Research PLC and Eastman Chemical^®^ formulated and patented a suspension of capsules based on the terpenes thymol and geraniol, as a postplanting nematicide under the name Cedroz^®^. In this study, we tested a wide range of different concentrations of a pure thymol on all developmental stages of a root-knot nematode (*M. javanica*). All these concentrations were tested not only for nematicidal effect but also for nematostatic effect and for vapor activity, comparing these to the contact activity.

Specifically, this study will focus on five key objectives:Assessment of the nematicidal and nematostatic activity of thymol on *M. javanica* J2s.Evaluation of thymol’s inhibition effect on undifferentiated eggs.Analysis of the hatch inhibition activity of thymol on egg masses.Examination of the contact and vapor activity of thymol in nematode-infested soil.Analysis of the effect of sublethal concentrations of thymol on the infectivity of *M. javanica*.

## 2. Results

### 2.1. Nematode Motility Bioassays

#### 2.1.1. Nematicidal Activity

The effect of thymol on the J2 motility of *M. javanica* is shown in Table 1. The percentage of immobile J2s increased with increases in the exposure period. Thymol showed more than 96% paralysis at concentrations of 1000 and 500 μL/L after exposure for 24 and 96 h, respectively. The results from the mobility in vitro bioassay showed that at the concentration of 250 μL/L, about 30% of J2s were paralyzed after 96 h of exposure. At concentrations of 125 and 62.5 μL/L, no significant effect on juvenile mortality was recorded. (Table 1).

#### 2.1.2. Nematostatic Activity

The present study found that there was no nematostatic effect observed. The permanent paralysis of J2s was monitored by keeping them in wells containing clean water for 12, 24, 48 and 96 h after evaluation. The percentage of paralyzed J2s was found to be similar to the results obtained from the previous experiment testing nematicidal activity using a bioassay.

#### 2.1.3. Effect of Thymol on Egg Development

Table 2 presents the inhibitory effect of thymol acting in different concentrations on the differentiation of eggs after exposure for 21 days. The lower percentage indicates higher efficacy at inhibiting egg differentiation (Table 2). A strong inhibition effect was recorded after comparing the control treatment (92.2%) with thymol at the concentrations of 250, 500 and 1000 μL/L (74.9%, 75%, 60.4%, respectively). Additionally, there was no significant difference between the treatment at a concentration of 62.5 and to the control treatment (Table 2). The experiment was terminated since no further egg differentiation was observed in the control treatment.

#### 2.1.4. Effect of Thymol on Egg Hatching from Egg Masses

The highest number of hatched J2s (about 85%) was recorded in egg masses that remained in clean water (control) throughout the experiment (Figure 2). Thymol at concentrations of 500 and 1000 μL/L exhibited the highest activity, inhibiting about 60% and 80% of egg hatching, respectively, compared to the control treatment. At concentrations of 62.5, 125, and 250 μL/L, thymol also resulted in more than 40% hatching reduction compared to the control treatment (Figure 2).

#### 2.1.5. Contact and Vapor Effect of Thymol against *M. javanica*

The toxic effects of thymol on the J2s were examined by the contact–vapor mortality in vitro bioassay (Figure 3 and Figure 4). It was observed that there was a corresponding increase in J2 mortality, as concentrations were increased from 62.5 to 1000 μL/L. Thymol at concentrations of 500 and 1000 μL/L displayed strong contact activity against *M. javanica*, inducing about 55% and 88% J2 mortality, respectively, at both temperatures tested. At the concentration of 250 μL/L, an almost 30% decrease in J2s in the soil was recorded in the contact bioassay, while the concentration of 62.5 μL/L showed the lowest effect on mortality, at both temperatures. On the contrary, the lack of vapor toxicity against J2s of *M. javanica* observed is irrelevant to the temperature.

#### 2.1.6. Sublethal Effect of Thymol on J2s Invasion (Pot Experiment)

Figure 5 presents the determined effect of the sublethal concentration activity of thymol in the in planta investigation against *M. javanica.* Thymol did not significantly decrease the number of females. Compared to the infested control, thymol treatments caused the lowest reduction in the nematode population in roots in all concentrations after 24 and 48 h of exposure (Figure 5). However, decreased numbers of females were recorded as the exposure time increased (96 h). The number of females was decreased to 38 and 36 as the concentrations were increased to 150 and 300 μL/L, respectively, after 96 h of exposure. No statistical difference was recorded for the number of females between the lower concentrations of thymol (37.5 and 75 μL/L) and the control treatment after 96 h of exposure.

## 3. Discussion

The root-knot nematode *M. javanica* causes severe damage to economically important crops such as tomato, with few effective chemical nematicides available. This study evaluate thymol’s strong nematicidal activity against *Meloidogyne javanica*; a percentage of about 95% of paralyzed J2s at a concentration of 500 μL/L after exposure for 96 h suggested that thymol has potential efficacy in terms of control. The motility and infectivity of *M. javanica* J2s was affected by the time of exposure as well as thymol concentrations throughout the experiment. Such results agree with those reported by [25], in which thymol paralyzed a high number of *M. javanica* J2s. Moreover, the authors of [35] previously reported a 57% mortality of *Meloidogyne incognita* J2s after exposure to thymol at concentration of 15 μg mL^−1^ for 48 h. Thymol was also documented for high in vitro nematicidal activity against *M. incognita* J2s [36].

To the best of our knowledge, this is the first report of thymol’s inhibition of egg differentiation using *Meloidogyne* species. Thymol showed, at a concentration of 1000 μL/L, a significant inhibition of egg differentiation for *M. javanica* eggs. In particular, thymol inhibited egg differentiation at the concentration of 1000 μL/L by about 35% after 21 days of incubation. It is important to test the efficacy of new molecules against eggs since the eggshell is the toughest part of nematode eggs and plays a role in their resistance to biological nematicides and chemicals [37].

In the egg hatching experiments, the egg hatching inhibition of *M. javanica* increased as the concentrations of thymol increased. There exists a direct relationship between the concentration of terpene and egg hatch inhibition activity. Hatching inhibition plays a significant role in plant protection strategies. As previously mentioned, the nematode eggshell consists of three layers and it is one of the most resistant biological structures known [37]. Thymol showed significant anti-hatching activity against *M. javanica* compared to the control treatment. A decrease of about 60% and 80% of juveniles hatched from egg masses was recorded after 35 days at 500 and 1000 μL/L concentrations, respectively. Substantial in vitro reduction in eggs hatching was observed even at low concentrations (250, 125, 62.5 μL/L). Oka et al. [34] reported that thymol inhibited the hatching of the root-knot nematode *M. javanica* at concentrations higher than 125 μL/L in in vitro experiments. Thymol had a significant inhibitory effect on *M. incognita* eggs at low concentrations (1, 2, and 4 mg/L) [31]. The eggshell structure of nematode eggs is different from the structure of the larval body wall, which might create differences in the ability of some natural active components to permeate through the eggshell. 

In our study, thymol exhibited strong contact mortality against *M. javanica* J2s in nematode-infested soil at high concentrations (500 and 1000 μL/L). However, thymol did not show significant fumigant-nematicidal activity, compared to the control. These findings are in agreement with our previous studies [23,38,39] in which carvacrol, geraniol and eugenol were shown to have the same behavior in soil as that reported in the present study for thymol.

Most of the juveniles that were treated with sublethal concentrations of thymol remained motile, although some of them were not able to locate and infect roots. Thymol at concentrations of 150 and 300 μL/L reduced the number of females per gram of root compared to the control, after exposure for 96 h (Figure 5). These results indicate that the duration of exposure and the concentration of the terpene could disorientate nematodes during root location [39]. However, no effect on the sublethal activity of thymol at low concentrations was detected (Figure 5). Oka et al. [34] reported that thymol, in *M. javanica*-infested sandy soil, reduced the root galling of cucumber seedlings at concentrations of 75 and 150 mg/kg.

## 4. Materials and Methods

### 4.1. Nematode Cultures

A population of *M. javanica* was reared on cv. Belladonna tomato seedlings (*Solanum lycopersicum* L.) in a greenhouse in the Agricultural University of Athens, Greece (25 ± 2 °C; 16 h of light and 8 h of darkness). Six-week-old tomato seedlings at the four-leaf stage were used for inoculation. Infested plants were kept for 40 days and then were uprooted, with galled roots being cut into 2 cm pieces after being gently washed free of soil. Sodium hypochlorite solution (1%) was used to collect eggs from roots [40]. Second-stage juveniles (J2s) were obtained by placing eggs on a Baermann funnel at the ambient temperature (27 ± 1 °C). The J2s used in the experiments were less than 2 days old.

### 4.2. Nematode Motility Bioassays

#### 4.2.1. Nematicidal Activity

Cellstar^®^ flat-bottom 24-well plates were used for all motility in vitro bioassays. The effect on the motility of J2s was tested by using solutions of thymol at the concentrations of 62.5, 125, 250, 500 and 1000 μL/L. Thymol (Merck; Darmstadt, Germany) was dissolved in ethanol (1%) (Sigma-Aldrich; Milano, Italy) to overcome the solubility problems and then serially diluted in distilled water containing 0.3% Tween-20 to prepare test solutions of the above concentrations. In all cases, final terpene solutions were prepared containing double the test concentration. Specifically, the thymol solution (0.5 mL) was pipetted into each well together with the J2 suspension (0.5 mL) at a ratio of 1:1 (*v*/*v*), to produce the desired concentration of the J2–terpene suspension. Preliminary tests showed that nematodes exposed to these concentrations of ethanol (1%) and Tween-20 (0.3%) were not affected. As a control we used distilled water. Approximately 40 J2s were used per treatment well in plates which were exposed to thymol solutions. All treated and control plates were covered with a lid to diminish terpene volatilization and incubated at 26 ± 1 °C. An inverted microscope (Zeiss; Oberkochen, Germany) was used to observe juveniles (100× magnification) after 12, 24, 48 and 96 h. All juveniles were categorized as either motile or paralyzed. The shape of the juveniles was found to be straight (I-shaped), bent (banana-shaped) and L-shaped. which indicated the difference between paralyzed (dead) or motile (alive) nematodes. For a window of 10 s, we checked the juveniles for motility by probing them with a needle. Lack of movement was considered a strong indication of paralysis. The experiment was conducted twice and all treatments were replicated five times.

#### 4.2.2. Nematostatic Activity

Thymol was dissolved in ethanol and diluted serially in distilled water containing Tween-20, resulting in solutions at concentrations of 62.5, 125, 250, 500 and 1000 μL/L. In total, 50 mL of each test solution was placed in a 250 mL Erlenmeyer flask and then newly hatched J2s were added. In all cases, working solutions were prepared containing double the test concentration and then mixed in flasks at a ratio of 1:1 (*v*/*v*) with a 50 mL suspension containing approximately 2000 J2s. To avoid a lack of oxygen due to the high number of nematodes in each flask, air was delivered using a plastic tube connected to an air pump. A cotton plug was used to cover the flask and all flasks were incubated at a temperature of 26 ± 1 °C in the dark. Distilled water was used as a control.

After 12 h, two solutions of 5 mL each were removed from every flask and used independently. The first solution was divided into five aliquots of 1 mL each and placed into wells (containing approximately 40 J2s per ml/well). J2s were categorized as either motile or paralyzed after being observed using an inverted microscope (100×). The second solution was sieved using a 38 μm sieve and rinsed 4 times using tap water. All J2s from the sieve were placed into a beaker and they were transferred to wells with approximately 30 J2s per well. To monitor recovery, J2s were counted using an inverted microscope (100×) after 12 h. The same procedure, as described above, was repeated after 24 and 48 h. The nematostatic effect was certified by observing if any J2s regained motility (non-permanent paralysis). Two solutions of 5 mL each were removed from flasks after 24, 48, and 96 h and the same procedure as that which was previously described was followed. The experiment was conducted twice and all treatments were replicated five times.

### 4.3. Effect of Thymol on Egg Development

Sodium hypochlorite solution (1%) was used to collect *M. javanica* eggs from infected tomato roots [40]. Eggs suspension was sieved through 53 and 38 μm sieves, rinsed thoroughly with tap water to become clearer and was collected into a 100 mL beaker. The number of eggs was quantified by using an inverted microscope (100×) and used directly in the bioassays.

The effect of thymol on egg development was tested at the concentrations of 62.5, 125, 250, 500 and 1000 μL/L. Ethanol and Tween-20 were used to overcome thymol insolubility in water as previously described. Distilled water was used as a control. Approximately 50 eggs, of which 90% were undifferentiated, were used per well (0.5 mL egg suspension), which were exposed to thymol solutions (0.5 mL) and incubated at 26 ± 1 °C. To prevent evaporation, we covered both treated and control plates with a lid.

The number of eggs developed and the number of J2s that emerged were counted every 7 days [41] using an inverted microscope (100×) (Zeiss, Germany). For monitoring egg development, eggs were observed on day 0 with the aid of an inverted microscope and were distinguished as either differentiated (fully developed juveniles) or undifferentiated (eggs containing only cells). Undifferentiated eggs were considered those with cell division (of one, two, or more cells).

After three weeks, the experiment was terminated since no further egg differentiation was observed in the control treatment. The experiment was conducted twice and each treatment was replicated four times.

### 4.4. Effect of Thymol on Egg Hatching from Egg Masses

Mature egg masses were collected from free-of-soil roots and placed in small 6 cm handmade plastic extraction trays. Each extracting tray was filled with thymol solutions at concentrations of 62.5, 125, 250, 500 and 1000 μL/L (Figure 6), after being dissolved and brought to the desired volume by using ethanol and Tween-20 in distilled water (as previously described). Egg masses maintained submerged in test solutions for seven days and then the test solutions were discarded. Thereafter, each egg mass was carefully submerged twice in clean water in order to remove excess thymol and was finally placed in an extracting tray filled with clean water. The extracting trays were covered to avoid a loss of water and placed in an incubator at 26 ± 1 °C. The number of hatched J2s was counted every seven days; then, they were discarded and the extracting trays were filled with fresh water. Emerged juveniles were counted every week for a period of five weeks when the experiment was terminated since J2s did not emerge any longer. At the end of the experiment, we smashed all the egg masses, we counted the unhatched eggs (per egg mass separately), and then we evaluated the percentage of the hatch inhibition activity of thymol on egg masses. The experiment was conducted twice and every treatment was replicated five times.

### 4.5. Contact and Vapor Effect of Thymol against M. javanica

Sandy soil was collected from a field (Gargalianoi village, Messinia, Southern Greece), and was sieved using a 2 mm aperture sieve. Then, it was sterilized for 20 min in an autoclave at 100 °C. Afterwards, the soil was oven-dried at 50 °C for 24 h to determine the maximum water-holding capacity (MWHC) (a gravimetrical measurement following the saturation of the soil with water, allowing it to drain for 24 h) [42].

Plastic pots (7 cm deep and 5 cm in diameter) were filled with 40 g of soil each and immediately inoculated with a nematode suspension of 500 J2s. In half of them, a plastic mesh net (of a size of 1 mm) was used to replace the removed plastic bottom. Thymol solutions were added to the pots with a plastic bottom (in vitro contact mortality bioassay), while plain water was added to the ones with the mesh net (in vitro vapor mortality bioassay). Subsequently, the pots were arranged in pairs, in a way that the pot with the mesh net was placed on top of the one with the plastic bottom (in vitro contact–vapor mortality bioassay) (Figure 7). To prevent moisture loss and minimize light effects, the pots were sealed with parafilm and covered with aluminum foil. The soil in the two pots did not make contact to avoid J2 migration. Moisture content was around 20% of the MWHC.

The efficacy of thymol against nematodes was tested at concentrations of 62.5, 125, 250, 500 and 1000 μL/L. A thymol stock solution was prepared in ethanol and Tween-20 (0.6%) to overcome insolubility, whereas distilled water with Tween-20 (0.6%) was used for further dilutions in accordance with the above method. Control pots consisted of soil with a certain moisture content and content of J2s. The experiment was conducted at either 20–22 °C or 30 °C. Plastic pots were kept in a climate room for three days. For the experiment at 30 °C, the plastic pots were placed on a metal tray with dimensions of 60 cm × 40 cm × 8 cm (L × W × H). A flexible silicone resistance connected to a thermostat was placed on the bottom surface of the metal tray. Wet sand was used to fill the metal tray on which plastic pots were placed throughout the experiment. Three days later, soil from every pot was totally removed and nematodes were extracted following a modification of Cobb’s decanting and sieving methods [43]. After two days at 26 ± 1 °C, the juveniles that had passed through the funnel were collected and counted using an inverted microscope at 100× magnification. The experiment was conducted twice while each treatment was replicated four times. The Abbot formula was used to calculate all numbers before being statistically analyzed [44]:$$\propto =\frac{\;\mathrm{mortality}\;\mathrm{treatment}-\mathrm{mortality}\;\mathrm{water}\;}{100-\mathrm{mortality}\;\mathrm{water}\;}\times 100\%$$

### 4.6. Sublethal Doses Effect of Thymol on Juvenile Invasion (Pot Experiment)

Tomato seedlings, of cv. Belladona, grown in plastic pots (of 10 cm deep and 6 cm in diameter) were used to evaluate in planta the efficacy of thymol against nematode invasion. All seedlings were 4 weeks old at the four-leaf stage. Four 250 mL Erlenmeyer flasks were filled with 100 mL solutions of thymol each at concentrations of 37.5, 75, 150 and 300 μL/L. Fifteen thousand newly hatched J2s were added into each flask and all flasks were incubated at 26 ± 1 °C. Ethanol (Sigma-Aldrich; Italy) and Tween-20 were used to produce test solutions of the above concentrations. Distilled water was used as a control. A total of 24 hours later, 1/3 of the suspension from every flask was transferred to a sieve (38 μm) and excess thymol was removed by gently washing it with tap water. Immediately, nematodes were transferred to the wells of a 24-well plate and they were scored as motile or paralyzed using an inverted microscope (Zeiss, Germany). The remaining J2s were kept in the flasks for another 48 h and 96 h when the same procedure as that which was previously described was repeated. A 5 mL solution from flasks containing 300 motile J2s of *M. javanica* was used to infect tomato plants (prepared as previously described) into every pot. All plants were placed in a growth room at 26 ± 2 °C and 30 days later they were uprooted. Roots were then carefully washed free of soil and stained with an acid fuchsin solution as described in [45]. These roots were washed in water and placed in vials containing equal volumes of glycerol and distilled water. The female nematodes were counted in the whole root system of each plant using a stereoscopic microscope at 12.5× magnification. The experiment was conducted twice in a randomized block design with five replicates per treatment while data were combined since no variation was revealed after preliminary statistical analysis.

### 4.7. Statistical Analysis

The data was analyzed using one-way ANOVA and the general linear model (GLM) to compare treatments. The results were then compared using the least significant difference (LSD) test. The analysis was performed using SAS statistical software (SAS University Edition). All experiments were conducted twice and they were combined and analyzed together since no variation was revealed between data.

## 5. Conclusions

In conclusion, this study indicates that the terpene thymol could be useful as a natural nematicide against *M. javanica*. Τhe nematicidal effect recorded in the experiments of the thymol was concentration-dependent, in conjunction with the increasing exposure time. At present, the discovery of new nematicides that are safe for human and non-target organisms is still a considerable challenge for nematode management. Focusing on the nematicidal activity of terpenes, the determination of the mode of action is essential for discovering friendly nematicides with formulations that provide efficacy at low concentrations, to minimize environmental risk.

## Figures and Tables

**Figure 1 plants-12-01851-f001:**
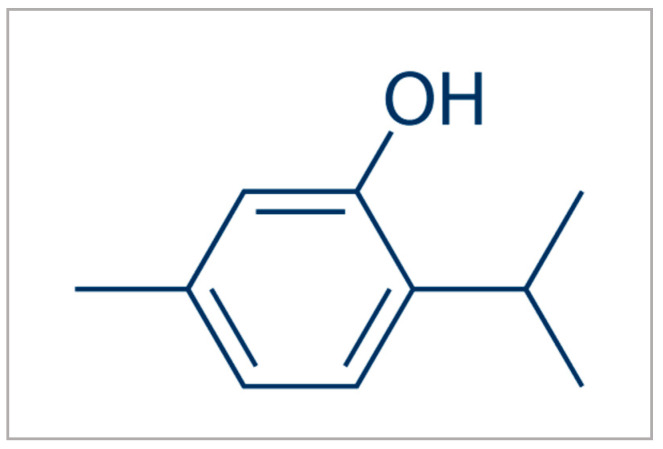
Molecular structure of thymol.

**Figure 2 plants-12-01851-f002:**
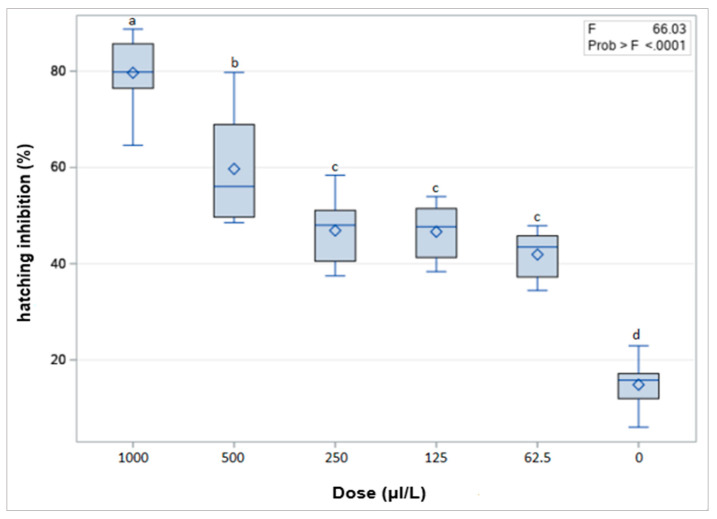
Effect of thymol on *Meloidogyne javanica* hatching (two experiments), after immersion of egg masses at the concentrations of 1000, 500, 250, 125, 62.5 and 0 μL/L for 35 days. Values are means of 5 replicates. Same lower-case letters indicate no significant difference according to the LSD test.

**Figure 3 plants-12-01851-f003:**
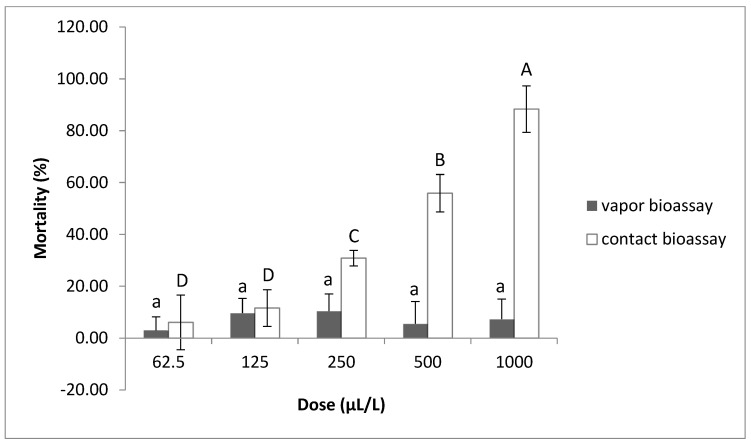
The mortality of *Meloidogyne javanica* J2s affected by the contact or vapor action of thymol in soil at the concentrations of 1000, 500, 250, 125 and 62.5 μL/L at 20–22 °C. Values are means of combined results from two experiments with 4 replicates each. Bars with the same letter indicate no significant differences according to the LSD test; upper-case letters refer to the contact bioassay while lower-case letters refer to the vapor bioassay.

**Figure 4 plants-12-01851-f004:**
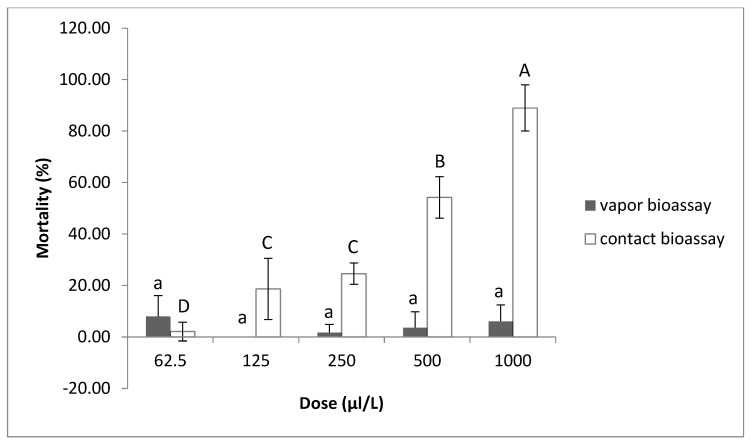
The mortality of *Meloidogyne javanica* J2s as effected by the contact or vapor action of thymol in soil at the concentrations of 1000, 500, 250, 125 and 62.5 μL/L at 30 °C. Values are means of combined results from two experiments with 4 replicates each. Bars with the same letter indicate no significant differences according to the LSD test; upper-case letters refer to the contact bioassay while lower-case letters refer to the vapor bioassay.

**Figure 5 plants-12-01851-f005:**
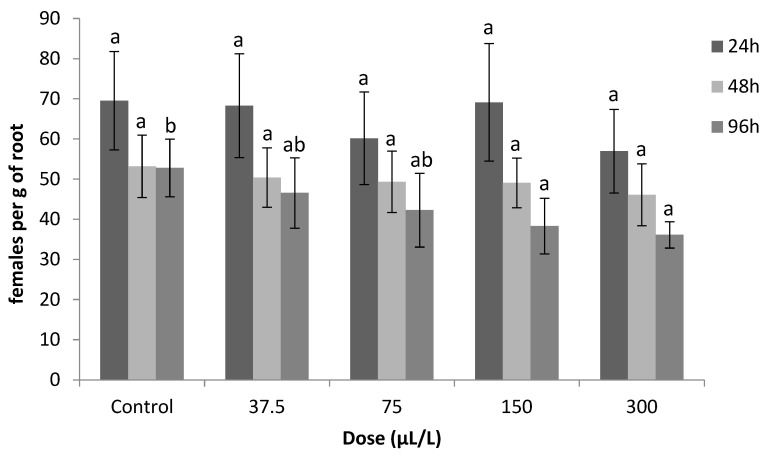
Numbers of females of *Meloidogyne javanica* per gram of root after immersion in thymol solutions at the concentrations of 0, 37.5, 75, 150 and 300 μL/L for 24, 48 and 96 h. Error bars represent the standard deviation of the mean (n = 5). The same letter in bars with the same color indicate no significant difference according to the LSD test (*p* < 0.001).

**Figure 6 plants-12-01851-f006:**
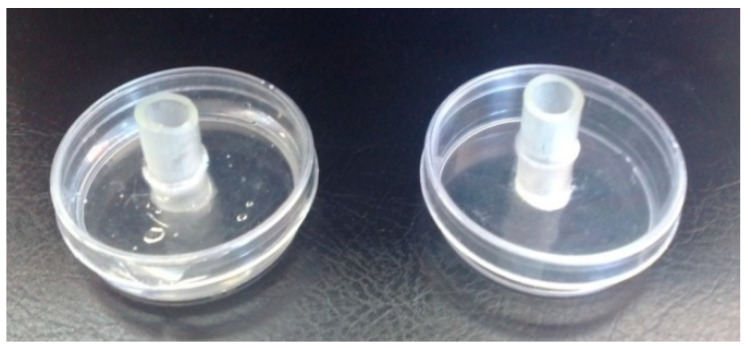
Egg masses placed in plastic extraction trays filled with terpene solutions.

**Figure 7 plants-12-01851-f007:**
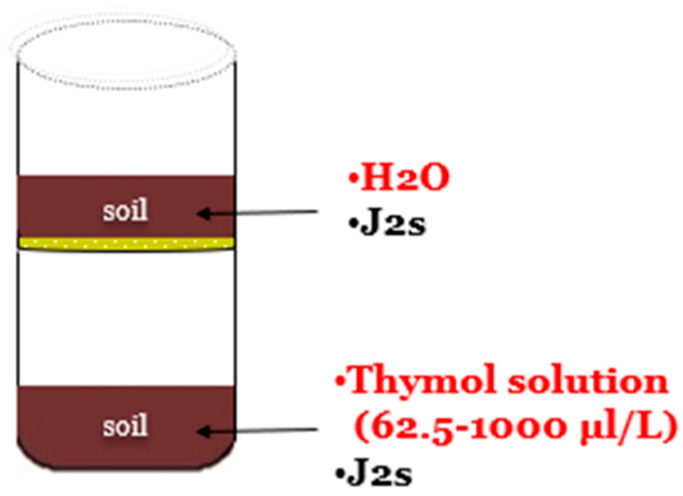
Pot trial methodology on the contact and vapor effect of thymol against *M. javanica*.

**Table 1 plants-12-01851-t001:** Effect of thymol on the motility of *Meloidogyne javanica* J2s after immersion in test solutions at the concentrations of 1000, 500, 250, 125, 62.5 and 0 μL/L for 12, 24, 48 and 96 h.

Dose (μL/L)	Exposure Time (hours)			
	12	24	48	96
	Paralyzed J2s (%)	Paralyzed J2s (%)	Paralyzed J2s (%)	Paralyzed J2s (%)
0	0 d	0.18 f	1.1 f	2.0 f
62.5	0.4 d	1.6 e	5.1 e	10.0 e
125	0.6 d	5.7 d	9.1 d	15.2 d
250	14.2 c	13.9 c	16.5 c	31.5 c
500	23.5 b	39.7 b	71.8 b	96.4 b
1000	81.2 a	98.0 a	100 a	100 a

Number of samples: 240; degrees of freedom: 24; number of replicates: 5. Values are means from two experiments. Values followed by the same letter in a column do not differ significantly according to LSD (*p* < 0.001).

**Table 2 plants-12-01851-t002:** Effect of thymol on the differentiation of *Meloidogyne javanica* eggs, after immersion of undifferentiated eggs at concentrations of 1000, 500, 250, 125, 62.5 and 0 μL/L.

Concentration (μL/L)	Exposure Time (21 Days)
	Eggs Differentiation (%)
0	92.2 a ^1^
62.5	84.1 ab
125	81.2 bc
250	74.9 c
500	75.0 c
1000	60.4 d

^1^ Values are means of combined results from two experiments with four replicates each, since no significant differences were observed after ANOVA using the data of both experiments. Values followed by the same letter in a column do not differ significantly according to LSD (*p* < 0.001).

## Data Availability

Suggested Data Availability Statements are available in section “MDPI Research Data Policies” at https://www.mdpi.com/ethics (accessed on 10 April 2023).

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
