# Peer review of "Nematicidal Potential of Thymol against *Meloidogyne javanica* (Treub) Chitwood"

_plants, 2023, doi:10.3390/plants12091851_

Round 1
Reviewer 1 Report (Previous Reviewer 1)
The authors have addressed the main comments to the previous submission.
Author Response
The manuscript has been revised.
Reviewer 2 Report (Previous Reviewer 3)
The ms "Nematicidial potential of Thymol against Meloidogyne javanica (Treub)Chitwood is actually provide additional experimental study in which Thymol an EO is evaluated for it nematicidial activity.
Although I find significant repetition to Oka et al study performed previously the additional experiment might provide some exposure and awareness for its potential application as alternative nematicidial cpmpond
There are some minor comments written throughout the ms which required some further reference

Author Response
A list of the amendments made to the manuscript according to the suggestions are listed below.

Reviewer 3 Report (New Reviewer)
This work aims to determine the Nematicidal Potential of Thymol Against Meloidogyne javanica . The nematicidal activity of thymol on root-knot nematodes was first reported in 1995 ( Soler-Serratosa et al 1995, Nematropica 26:57-71). Since then, several studies have determined the nematicidal potential of this monoterpene, which is also the main active component of the oil extracted from different plant species of the genus Thymus, Ocimum, Origanum, Satureja and many other essential oils.
Specifically, they have already been determined:
- - in vitro nematicidal effects on juveniles of Meloidogyne incognita and M. javanica, (Oka et al, 2000, Ibraim et al, 2006, Ntalli et al 2010, Andres et al, 2012)
- - Effects on the inhibition of egg hatching in Meloidogyne incognita and M. javanica (Oka et al, 2000, Ibraim et al, 2006, Andres et al, 2012)
- - In vivo nematicidal effects on tomato plants. (Soler- Serratosa et al 1995, Oka et al 2000, Rasoul an Mona, 2013)
- - Its potential role as a disruptor in the chemotactic-infective behavior of root-knot nematodes (J2) has been investigated.(Kihika et al 2017)
-
Likewise, as mentioned in the manuscript (last paragraph of the introduction) there are currently several nematicidal products with thymol as an active ingredient on the market.
In the PDF manuscript some corrections and suggestions are marked

Author Response
A list of the amendments made to the manuscript according to the suggestions are listed below.

Reviewer 4 Report (New Reviewer)
1. Lines 238-239. What does it mean "disturbing"the juvenile?
2. Lines 251-253. How is it feasible to have 40J2s per ml? The flask contained 1,200 J2s in a total volume of 100ml, which means 12 J2s/ml.
3. Line 324. The incubation at 20-22 C was in an incubator or at room temperature?
4. Lines 112-117. How the percentage of hatching was estimated? Did you estimate the total number of eggs per egg mass?
5. There are a few citations of papers with the effect of thymol against Meloidogyne. Did they use similar commercial products (e.g Merck) like the one used in this study?
Author Response
A list of the amendments made to the manuscript according to the suggestions are listed below.

Round 2
Reviewer 3 Report (New Reviewer)
First of all, I want to express my appreciation for the effort that the authors of the manuscript have made in answering my comments, however, I do not agree with their answers at all. I emphasize once again that the nematicidal potential of thymol against root-knot nematodes has been widely demonstrated and that there are nematicidal products formulated with thymol on the market.
1. In this work, unjustified experimental variations (in times, doses, effects on egg differentiation) of standardized bioassays validated by numerous publications are carried out, which do not provide new or significant knowledge regarding the PREVIOUSLY KNOWN NEMATICIDAL ACTIVITY OF THYMOL (on J2, on egg masses, in the infection of tomato plants).
2. The median lethal dose, or LD50, is used as a measure of the lethal dose of a compound. Specifically, the LD50 represents the dose at which a substance is lethal to 50% of the pathogenic inoculum. This value is used as an indicator of the relative toxicity of a compound and, in this case, of the nematicidal potential of thymol.
3. The LD50 does not induce the death of the entire population of nematodes, it is only lethal for 50% of the inoculum, so it is the most appropriate dose to use in this experiment.
4. Bioassays for in vitro nematicidal activity from numerous published studies demonstrate the robustness of the results obtained when performed with an inoculum of approximately 100 juveniles per well (in 96-well plates with a volume of 100 µl/well).
5. Perhaps my comment was misinterpreted. It is correct and recommendable to apply Abbot's formula to express mortality results. But these corrected data with Abbot's formula cannot be subsequently analyzed and compared with those of the control treatment by ANOVA and LSD.
6. In this study the thymol compound has not been tested on at least one other genus of nematodes or on other plant hosts. The proposed objectives and the results obtained do not provide new knowledge.
Author Response
The reviewer 3 "emphasizes once again that the nematicidal potential of thymol against root-knot nematodes has been widely demonstrated and that there are nematicidal products formulated with thymol on the market", however the few registered products in the market contain not only thymol but also other terpenes or numerous additives which is a totally different case study comparing to ours.
1. I agree with reviewer 3 that there are numerous publications on the same subject but:
a. there is not any publication in the literature to test thymol in all developmental stages of root-knot nematodes (M. javanica in our case) and also in the same way as we did.
b. The reviewer 3 is confused since it is different to run experiments using thymol (pure substance) and it is totally different case to run experiments using oil containing thymol and many other substances.
c. I don't think that the reviewer 3 has got the privilege to judge if this piece of work is new or SIGNIFICANT. All the experiments presented are scientifically correct, using the appropriate materials and methods and presenting new knowledge (in agreement or disagreement with other findings of previous researchers) to the academic community. Also the other 3 reviewers agreed this piece of work to be published.
2. The LD50 is relevant to: a) the species of the nematode, b) the developmental stage of the nematode, c) the time, d) the origin of the product. That is why you cannot use the LD50 of one researcher and find the same results with him. This is the main reason why we have used a series of concentrations to test thymol on M. javanica. If you use only the LD50 you get much lower information instead of using a series of concentrations as we did.
3. Again, reviewer 3 has not the privilege to advise the authors what is the most appropriate dose to use in their experiments. The experiments usually are designed and conducted by the authors and the reviewer is responsible to review the manuscript.
4. This point made by the reviewer 3 is quite arbitrary. In our opinion (after many published papers using terpenes, essential oils and chemicals) 100 J2s in a small quantity of 100 μl per well is quite high. If you keep 100 J2s for 96 hours in a well then many of them will be dead due to lack of oxygen. Also when J2s are overcrowded in a small area then the results are not legitimate.
5. There is no control data presented in Figures 3 & 4, in the experiment "Contact and vapor effect of thymol against M. javanica" where we have used the "Abbott formula". We are asking the reviewer 3 to pay more attention studying the manuscript and interpreting the data presented.
6. The final statement made by the reviewer 3 is arbitrary. The Associate Editor or the Editor should decide if this manuscript offers new insight in the scientific community.

This manuscript is a resubmission of an earlier submission. The following is a list of the peer review reports and author responses from that submission.
Round 1
Reviewer 1 Report
The authors describe the activity of the terpenoid thymol on various parameters of M. javanica mortality and life cycle. I congratulate the authors on the work performed. Nevertheless, although the work is well described, in some sections the English language is confusing, I would strongly recommend proofreading by a native English speaker with a scientific background. Also, I believe that you can show the final results after statistical treatment without showing the results of two equal experiments, which seems unnecessary.
I suggest some changes throughout the document (using the document line numbering in the latest file).
Line 11 – Change “cause severe economic damage to agricultural crops” to “cause severe damage to agricultural crops”
Line 13 – I am more accustomed to express concentration through mg/L or similar units than with ppm. Please change throughout the manuscript text.
Line 17 – change “more than 58%” with the exact number. How much more than 58%?
Line 20 – change “presenting” with “presented”
Line 23 – delete the “-“
Line 26 – delete “most” or “major”
Line 28 – equivalent to billions of what? please specify.
Line 31 – Which process? Please be clear in your affirmations.
Line 52 – change “terpenenoids” to “terpenoids”.
Line 54 – change “which are the main” to “which can be the main”, the EOs of a considerable number of plants are not dominated by terpenes.
Line 60 – please provide a figure with higher definition or quality.
Line 69 – Is “differentiation of thymol” the accurate term? What do you mean by differentiation? Is it the inhibition of egg differentiation?
Line 82 to 84 – Perhaps this part is more adequately placed in the material and methods section.
Line 99 to 100 – Please explain this affirmation in a clearer manner. Did you test different number of eggs at each concentration? Also, did you not test egg hatching in this parameter? Where are the results? Please explain clearly what you mean with “differentiation”.
Line 159 – please check if the letters for significance are correctly placed.
Line 164 – Your discussion is not very explanatory and mostly descriptive. Perhaps you could suggest some of the biochemical mechanisms behind thymol activity on the nematode. What is the common effects of thymol biocidal activity and can it relate to the composition of RKN cuticle, for example.
Line 214 – The first sentence is unnecessary, please delete.
Line 225 to 240 – This section is confusing. I would advise to rewrite in a clearer manner. Also, what volume of nematode solution did you use? You mentioned that you had terpene solutions with double the concentration required. Was that because you used an equal volume of nematode solution?
Line 242 to 244 – You repeat the methodology of obtaining solutions in every parameter. I would suggest you write an initial section were you describe everything related to the way you obtained the thymol solutions. Then you won’t need to repeat.
Line 299 – I had difficulty following the experimental procedure for this section. I think you would benefit from an explanatory figure, displaying the stepwise protocol.
Reviewer 2 Report
The manuscript presents information on the nematicidal potencial of thymol against M. javanica, very interesting topic for the development of new strategies for the sustainable management of phytonematodes.
Several experiments were performed, although, most of the data confirm results previously reported for both, thymol and M. javanica. Therefore, I think that the manuscript, in the current form, does not carry enough scientific news.
However, I would suggest the authors to deepen interesting points of the manuscript that would bring new knowledge, such as: the effect of Tymol on the embryo development, for example, through microscopy techniques, inhibition mechanism, ect. and/or the efficiency of the use of tymol for the control of root knot nematodes in crops, for example, through greenhouse and field assays, formulation, effects on plant growth.
Reviewer 4 Report
This manuscript describes experiments performed to assess the response of the root-knot nematode species Meloidogyne javanica to thymol. This work is important because of the few management options for economical and environmentally sustainable control of plant-parasitic nematodes in agricultural production. The work would be of interest to nematologists and plant protection specialists, and also perhaps those studying the intersection of microbes and soil chemistry.
However, it is unclear how this manuscript is a significant advancement and uniquely different from the work reported by Oka et al. 2000. Phytopathology 90:710-715, who also described the identification and reaction M. javanica to thymol. Oka et al. 2000 identified thymol as having nematicidal properties against M. javanica and also reported on work towards identifying concentrations needed to achieve immobility of J2 and inhibit hatching of eggs.
Further, in the present manuscript, several of the experiments are confusing in their objectives and hypotheses being tested. It is not quite made clear what the difference is between the hatching from egg masses experiment and the egg differentiation experiment. Also, in the pot experiment regarding the sublethal doses of thymol on J2 invasion of the roots, what is being tested in this experiment? It appears that J2 were exposed to varying concentrations of thymol, but then removed from the solution and inoculated onto plants, but at an equal population level of 300 J2 nematodes? Figure 5 does not appear to show any significant or interesting trends?
Lastly, the language and writing throughout the manuscript is a bit confusing and could benefit from review by a professional language editing service.
Thank you for the opportunity to review this manuscript.
Additional specific comments:
Title: "javanica" should not be capitalized in the title
Line 31: This sentence is quite confusing, what process are you referring to and what do these regulations mean?
The text in Figure 2 is small and the figure is generally difficult to read. Figure 2 appears to be in a different format than the other figures?
Line 238: around this line, it is unclear if the ethanol-only and Tween-only controls were included in the experiment. These controls are important and should be included.
In section 4.5, it is difficult to visualize how the vapor contact experiment was performed. Perhaps a photo of the apparatus setup or a diagram could be included?
In section 4.7, additional details regarding the statistical analysis are needed. Please spell out "LSD" at first mention. It is mentioned that datasets from two replicate experiments were combined if "no variation was revealed between data" - what kind of variation? What tests were performed to determine this?